# Detection and Characterization of *Leptospira* Infection and Exposure in Rats on the Caribbean Island of Saint Kitts

**DOI:** 10.3390/ani10020350

**Published:** 2020-02-22

**Authors:** Sreekumari Rajeev, Kanae Shiokawa, Alejandro Llanes, Malavika Rajeev, Carlos Mario Restrepo, Raymond Chin, Eymi Cedeño, Esteban Ellis

**Affiliations:** 1Ross University School of Veterinary Medicine, Saint Kitts, KN 0101, West Indies; bluerose_lotus@yahoo.co.jp; 2Instituto de Investigaciones Científicas y Servicios de Alta Tecnología (INDICASAT AIP), Panama City 0801, Panama; allanes@indicasat.org.pa (A.L.); crestrepo@indicasat.org.pa (C.M.R.); rchin3@u.rochester.edu (R.C.); eymicedeno@gmail.com (E.C.); ellis.esteban@gmail.com (E.E.); 3Department of Ecology and Evolutionary Biology, Princeton University, Princeton, NJ 08544, USA; mrajeev@princeton.edu

**Keywords:** *Leptospira*, rats, Caribbean, detection, mixed infections, immune response

## Abstract

**Simple Summary:**

Leptospirosis is a widespread zoonotic and potentially life-threatening disease in humans and animals. Many animal species maintain *Leptospira*, the bacterial agent causing this disease, in the kidneys and they shed the bacteria in urine. These animals act as a source of infection and environmental contamination. *Leptospira* infection has been previously reported in several animals on the Caribbean island of Saint Kitts, yet, no data is currently available on rats, a significant reservoir host of this pathogen. The main goal of this study was to detect *Leptospira* infection and exposure in two species of rats on Saint Kitts, by using a complementary set of diagnostic tools. Infecting *Leptospira* strains were subsequently characterized with a combination of serologic, molecular, and genomic techniques. Results show a relatively high prevalence of infection with *L. interrogans* (serogroup Icterohaemorrhagiae) and *L. borgpetersenii* (serogroup Ballum), with evidence supporting mixed infection with both species in some rats. Our study suggests the use of multiple diagnostic tests to enhance the results of *Leptospira* surveillance studies and diagnostic investigations.

**Abstract:**

In this study, we detected and characterized *Leptospira* infection and exposure in rats on the Caribbean island of Saint Kitts for the first time. We detected *Leptospira* infection in 17/29 (59%), 14/29 (48)%, and 11/29 (38)% of rats by RT-PCR, culture, and immunofluorescence assay, respectively. Whole genome sequencing (WGS) and analysis and serogrouping of 17 *Leptospira* strains isolated from rats revealed their close relationship with *L. interrogans* serogroup Icterohaemorrhagiae (*n* = 10) and *L. borgpetersenii* serogroup Ballum (*n* = 7). WGS, serogrouping, and additional PCR tests on rat kidneys confirmed mixed infections with *L. interrogans* and *L. borgpetersenii* in the kidneys of three rats. Microscopic agglutination test (MAT) was positive for 25/29 (87%) of the rats tested, and the response was restricted to serovars Icterohaemorrhagiae {24/29(83%)}, Mankarso {4/29(14%)}, Copenhageni {4/29(14%)}, Grippotyphosa {2/29(7%)}, and Wolffi {1/29(3%)}. Interestingly, there was no agglutinating antibody response to serovar Ballum. We observed a similar pattern in the serologic response using *Leptospira* isolates obtained from this study with each of the rat sera, with strong response to *L. interrogans* isolates but minimal reactivity to *L. borgpetersenii* isolates. Our findings suggest the use of multiple complementary diagnostic tests for *Leptospira* surveillance and diagnosis to improve the accuracy of the data.

## 1. Introduction

Leptospirosis can cause life-threatening disease in animals and humans, and cases are on the rise in subtropical and tropical regions, where climatic, socioeconomic, and environmental factors collide to create high-risk populations [1,2]. Pathogenic *Leptospira* colonizes renal tubules of a wide array of animal hosts, resulting in shedding in the urine. Transmission to humans and animals is often due to direct or indirect contact with infected animal reservoirs or contaminated environments, such as water, soil, or mud [2,3]. Infection in incidental or accidental hosts with non-host adapted serovars, usually acquired from a contaminated environment, may result in mild to severe clinical disease that potentially leads to life-threatening renal, hepatic, and pulmonary disease [2,3]. In contrast, in the maintenance or reservoir host species, such as rodents or livestock, infection with host-adapted strains is often perpetuated through direct contact between animals, resulting in chronic, persistent, and asymptomatic colonization of renal tubules and the reproductive tract [2]. In livestock species, chronic *Leptospira* infection can result in economic consequences due to reproductive loss, in addition to the sporadic clinical disease presentations observed in other domestic animals [2]. Soon after the discovery of *Leptospira* infection in humans, rats were identified as one of the key reservoirs and risk factors for leptospirosis [4]. Proximity to infected rodents is a documented risk factor for acquiring *Leptospira* infection in humans and animals in both urban and rural settings [5]. A recent literature review we conducted identified geographic and species-level variations in *Leptospira* prevalence among rats [6]. Our report also emphasized the need for enhanced surveillance programs using standardized methods. 

Estimated annual human mortality and morbidity in the Caribbean region are higher compared to other Global Burden of Disease regions [1]. In Saint Kitts, a small island located in the Caribbean region, human mortality (5.87) and morbidity (107.87) rates per 100,000 population due to *Leptospira* infections are estimated [1]. Recently, we reported infection and exposure to a diverse set of *Leptospira* serovars in multiple animal species on the island [7,8,9,10,11]. Yet, there is no data on *Leptospira* infection in one of the major global reservoir hosts, rats. In this study, we detected and characterized *Leptospira* infection and the immune response in two rat species on the island of Saint Kitts, using a combination of diagnostic tests.

## 2. Materials and Methods 

### 2.1. Ethical Statement

We used animal sampling protocols approved by the Ross University School of Veterinary Medicine (RUSVM) Institutional Animal Care and Use Committee (IACUC) ethical approval code # 17-01-04).

### 2.2. Study Location and Animals

Saint Kitts is a small, 67.18 sq. miles, Caribbean island located in the Lesser Antilles region (longitude 17.3434° N and latitude −62.7559° W). Considering the animal *Leptospira* seroprevalence rates on the island, ranging from 4% in cats to 80% in cattle [8,9], we set a sample size of a minimum of 9 rats at a 99% confidence interval in an infinite population with an assumed prevalence rate of 40% [12]. Sherman and Tomahawk traps were set up overnight at randomly selected sites using monkey chow as bait. Traps were placed in the late evening and routinely checked early morning. We captured rodents from three randomly selected sites, transported them to the RUSVM necropsy facility, and euthanized them using carbon dioxide gas. We collected blood via cardiac puncture immediately after euthanasia, separated the serum samples by centrifugation, and stored them in a −80 °C freezer. We performed a complete necropsy on each of the animals captured and recorded the date and location of trapping, species, sex, body weight, body/tail length, sexual maturity, pregnancy status, and presence of wounds or any other significant gross lesions. We performed culture and direct immunofluorescence assay (DFA) on kidney and urine on the same day of collection and stored the leftover samples in a −80 °C freezer for DNA extraction and subsequent serological tests. The rat species were initially determined based on phenotypic characteristics and external measurements and later confirmed by mitochondrial cytochrome *b* (*mytC*) amplification and sequencing [13].

### 2.3. Leptospira Culture 

We used a standard *Leptospira* culture procedure with modifications [14]. Briefly, a portion of the kidney was homogenized by passing through a 1-mL tuberculin syringe (4010-200V0, Norm-Ject^®^, Henke-Sass Wolf, Nörten-Hardenberg, Germany) and further suspended in 1.5 mL of sterile water. The supernatant (500 μL) was inoculated into liquid Ellinghausen-McCullough-Johnson-Harris (EMJH) media (Royal Tropical Institute, KIT Biomedical Research, The Netherlands) with 0.01% 5-fluorouracil and incubated at 29 °C for up to 7 months. Urine, if available, was also inoculated in EMJH media. The cultures were examined by dark field microscopy (DFM) for contamination within 24 h. Uncontaminated samples (500 μL) were subcultured into a new EMJH media (4.5 mL). If contamination was observed, the entire media was filtered through a 0.22-μm filter with 30 mL of fresh EMJH media. Cultures were checked for *Leptospira* growth every two weeks using dark field microscopy (DFM), and positive samples were subcultured and maintained in EMJH media supplemented with *Leptospira* Enrichment (Becton, Dickinson and Company, Sparks, MD, USA).

### 2.4. Leptospira Detection

We used DFA and real-time polymerase chain reaction (RT-PCR) for *Leptospira* detection from rat samples. For DFA, the homogenized kidney supernatant (5 μL) was spread on a glass slide with 10-mm circles, dried, fixed in chilled acetone, and incubated at 37 °C for 1.5 h after adding fluorescein isothiocyanate conjugated polyclonal anti-*Leptospira* antibody (National Veterinary Services Laboratory, IA, USA). The slides were washed three times in PBS and observed under 20 × objective of a UV microscope after placing mounting media and a coverslip. Samples were recorded as positive when fluorescent organisms with morphology compatible with *Leptospira* were observed. 

For RT-PCR, DNA was extracted from the supernatant of homogenized kidney samples using DNeasy Blood & Tissue Kits (QIAGEN Scientific Inc., MD, USA), following the manufacturer’s protocol. RT-PCR targeting *lipL32* was performed on a Smart Cycler (Cepheid Inc., CA, USA) as described previously [15]. 

### 2.5. Detection of Host Antibody Response

We used the microscopic agglutination test (MAT) to evaluate the humoral immune response to *Leptospira*. MAT was performed with a 21 *Leptospira* serovar panel, as previously described [8]. We screened sera against each of the serovars in the panel at a final dilution of 1:50, and samples showing 50% agglutination were recorded as positive. Titers were determined for positive samples. To assess the humoral immune response in individual rats, to the isolates obtained in this study, we conducted a cross-isolate MAT by treating each of the individual rat sera to all of the isolates obtained in this study. We used qualitative grading of agglutination activity to assess the intensity of the immune response.

### 2.6. Characterization of Leptospira Isolates

We completed whole genome sequencing (WGS) of *Leptospira* isolates using the Illumina platform. The passage number of the isolates at the time of DNA extraction ranged from 6–17 depending on the density and purity of the culture. DNA was extracted using the MasterPure™ DNA purification kit (Epicentre, WI, USA) and sequencing libraries were prepared with the DNA library prep kit for Illumina (New England Biolabs, MA, USA), following manufacturer’s instructions. Fragments were sequenced in a HiSeq 2500 instrument (Illumina, CA, USA), resulting in an average of ~4 million 150-bp paired-end reads per sample. BLAST [16] was used to align the reads from each isolate against the Genbank database. Reads were assembled de novo with SPAdes (version 3.12.0) [17] and the assembled genomes were further annotated by using the Prokaryotic Genome Annotation Pipeline (PGAP) [18]. Raw WGS reads and annotated genomes were deposited on the databases of the National Center for Biotechnology Information (NCBI) under BioProject PRJNA543109.

Genetic variability among the isolates was assessed by first aligning the Illumina reads from each isolate against the reference genome for the corresponding serogroup, namely, *L. interrogans* serovar Copenhageni strain Fiocruz L1-130 (BioProject PRJNA10687) and *L. borgpetersenii* serovar Castellonis strain 200801910 (BioProject PRJNA74139), by using BWA (version 0.7.12) [19]. Small variants, including single nucleotide polymorphisms (SNPs) and insertion/deletions up to 3 bp in length, were identified from read alignments with the Genome Analysis Toolkit (GATK) (version 3.5) [20].

In order to confirm the presence of mixed infections with two *Leptospira* species, we performed conventional PCR reactions targeting insertion sequences (ISs) from isolate-positive kidneys and isolates as described previously [21]. We also pursued a presumptive serogroup identification by treating each of the isolates with a panel of polyclonal rabbit anti-*Leptospira* sera (Royal Tropical Institute, KIT Biomedical Research, The Netherlands) against each of the 21 strains used in our MAT panel.

### 2.7. Statistical Analysis 

We used GraphPad Prism (version 8.3.0) (GraphPad Software, La Jolla CA, USA) to estimate prevalence and confidence intervals. We used R (version 3.5.1, R Core Team, 2018) to analyze the results of our cross-isolate assay, using a generalized linear mixed effects model with MAT response as the (binomial) response variable, isolate species, host species, host maturity, and whether the isolate originated from the host tested as fixed effects, and the individual rat as a random effect.

## 3. Results

### 3.1. Detection of Leptospira Exposure and Infection in Rats

We captured 29 rats (labeled R1–R29) from January to July 2017. The location of trapping is given in Appendix A. Phenotypic characteristics and sequencing of the *mytC* gene confirmed rodent species captured as *Rattus norvegicus* (*n* = 18) and *Rattus rattus* (*n =* 11). Notable differences were observed in the results obtained with the three methods used to detect *Leptospira* infection in rats. Globally, we detected *Leptospira* renal colonization in 59%, 48%, and 38% of rats tested by RT-PCR, culture, and DFA, respectively (Table 1). Observation of organisms in DFA-positive samples was explicit, showing numerous fluorescent cells with morphology compatible with *Leptospira* (Appendix A). 

### 3.2. Characterization of Leptospira Isolates

We obtained 17 *Leptospira* isolates from 14 rats, respectively labeled after the individual rat from which they were isolated (Table 2). All isolates were obtained from kidney samples, except for those labeled with an ‘L’, which were obtained from the urine. By conventional serogrouping, 10 isolates were identified as serogroup Icterohaemorrhagiae and eight isolates as serogroup Ballum. One of the isolates (R14L) reacted strongly to both anti-Icterohaemorrhagiae and anti-Ballum antiserum, thus suggesting the coexistence of both serogroups in the culture. Among serogroup Icterohaemorrhagiae isolates, we also observed mild to moderate cross-reactivity with anti-Canicola antiserum (*n* = 3). Among serogroup Ballum isolates, two also reacted with anti-Alexi serum (*n* = 2). 

BLAST searches of the WGS reads against the Genbank database identified all the isolates in serogroup Icterohaemorrhagiae (*n* = 10) as *L. interrogans* and those in serogroup Ballum (*n* = 7) as *L. borgpetersenii*. Globally, the highest percentage of sequence reads from *L. interrogans* isolates matched to sequences from serovar Copenhageni (serogroup Icterohaemorrhagiae), while those from *L. borgpetersenii* isolates matched to sequences from serovar Castellonis (serogroup Ballum).

In isolates R6, R6L, R14, R14L, R28, and R29, primarily matching *L. borgpetersenii* sequences, we observed a relatively low but noticeable number of reads matching *L. interrogans* sequences, thus suggesting mixed infections. The most remarkable case is that of isolate R14L, with 83% of reads matching *L. borgpetersenii* and the remaining 17% matching *L. interrogans*. De novo assembly of the Illumina reads from this isolate resulted in proper separation of the genomes of the two co-infecting species. However, to avoid further confusion, only the genome of the species with the higher percentage of reads in the sample (*L. borgpetersenii*) was submitted to the Genbank database.

In order to confirm the presence of the two *Leptospira* species in the isolates mentioned above, we used PCR amplification of insertion sequences IS1500 and IS1533, which can differentiate *L. interrogans* from *L. borgpetersenii*. To rule out the possibility of contamination, we also compared the cultures of these *Leptospira* isolates with the original kidney samples from the corresponding rats. We obtained evidence of mixed infection in the kidneys of three rats, namely, R14, R28, and R29 (Appendix A). Likewise, consensus between both markers confirmed the presence of *L. interrogans* and *L. borgpetersenii* DNA in isolates R14, R14L, R28, and R29. 

We also assessed and compared the genetic variability among all the isolates by mapping the reads from each isolate against the reference genome for the corresponding serogroup. We found relatively low genetic variability among isolates, measured in terms of SNP and small indels (Appendix A). All isolates within each serogroup have roughly the same set of shared variants. On average, *L. interrogans* isolates have 40 variants located within coding sequences, the vast majority of which affect genes associated with mobile elements, coding for the transposase or integrase components. In *L. borgpetersenii* isolates, the number of variants affecting genes is slightly higher, 90 on average, due to the higher number of transposable elements present the genome of this species [22].

### 3.3. Characterization of Immune Response in Rats

We used MAT to evaluate the humoral immune response to *Leptospira* infection in sampled rats. A relatively high proportion of rats {25/29 (87%)} were MAT-positive. Agglutinating antibody response was detected in 5 out of 21 serovars used in this study. Positive MAT response was observed for serovars Icterohaemorrhagiae in {24/29 (83%)}, Mankarso{4/29(14.%)}, Copenhageni {4/29(14%)}, Grippotyphosa {2/29(7%)}, and Wolffi {1/29(3%)} (Figure 1). The MAT titers ranged from 50–1600. Interestingly, we did not observe any MAT response to serovar Ballum.

We also conducted a cross-isolate MAT to assess the agglutinating antibody response in each of the individual rats to the isolates obtained in this study (Figure 2). We measured the levels of the immune response using the intensity of agglutinating activity with isolates and individual rat sera. MAT response varied by the individual rat, with some rats exhibiting high responses and others that tended to have overall low or no response to any isolate. However, across all individuals, very few responded to *L. borgpetersenii* when compared to *L. interrogans* isolates, regardless of rat species (odds ratio (OR) = 0.03, 95%CI: 0.001–0.1, *p* < 0. 001) In addition, the sera from rats suspected to have mixed infection with both species reacted mostly with *L. interrogans* isolates in this assay.

## 4. Discussion

In this study, we confirmed for the first time *Leptospira* infection and immune response in rats inhabiting the Caribbean island of Saint Kitts. We also characterized the infecting *Leptospira* spp., reported the presence of mixed infection, and highlighted an absence of agglutinating antibody response to one of the infecting serogroups.

Previously, we reported a significant variation in *Leptospira* diversity and geographic prevalence in rats worldwide [6]. The most frequent *Leptospira* serovars reported per geographical regions are Canicola in Africa, Icterohaemorrhagiae and Sejroe in Europe, and Icterohaemorrhagiae in Asia, North America, South America, and the Caribbean. The serovars belonging to the serogroup Ballum (Ballum, Arborea, and Castellonis) were also reported in various parts of the world. *Leptospira* exposure and infection is common among animals on the island [7,8,9,10,11]. The highest seroprevalence was reported in cattle (80%) followed by dogs (73%), pigs (69%), African green monkeys (49%), sheep (35%), goats (24%), mongoose (8%), and cats (4%). The *Leptospira* isolates from rats in this study were limited to serogroup Icterohaemorrhagiae and Ballum; however, we have documented MAT response to a variety of other serovars in animals on the island, suggesting the role of other animal species contributing to *Leptospira* epidemiology. There is widespread exposure to the members of the serogroup Icterohaemorrhagiae in animals on the island but exposure to serogroup Ballum is predominantly documented only in monkeys and cattle. Vaccination is not practiced in animals except for dogs. There is a scarcity of data on current human exposure and incidence of clinical leptospirosis. 

MAT is a serological technique widely used to asses *Leptospira* seroprevalence globally. MAT uses live *Leptospira* antigens and is a reference test to compare other serologic methods [23]. Since antibodies against *Leptospira* do not cross-react with other bacteria, the technique has acceptable specificity in detecting anti-*Leptospira* antibodies. Nevertheless, there is apparent cross-reactivity between serovars within the same serogroup and sometimes even among unrelated serovars, a phenomenon described as paradoxical reactions, rendering reduced specificity for this test to detect infecting serovars [24]. In our study, of the 21 serovars used in MAT, three were members of serogroup Icterohaemorrhagiae and one of serogroup Ballum, the infecting serogroups identified in Saint Kitts rats. The rat sera agglutinated with members of the serogroup Icterohaemorrhagiae variably, while not agglutinating with serovar Ballum. *L. interrogans* and *L. borgpetersenii* isolates from this study were genetically similar to serovars Copenhageni (serogroup Icterohaemorrhagiae) and Castellonis (serogroup Ballum), respectively.

A higher percentage of rats were positive for agglutinating antibodies to serovar Icterohaemorrhagiae, not serovar Copenhageni, the serovar to which all *L. interrogans* isolates from Saint Kitts were more closely related. Since both of these serovars belong to the serogroup Icterohaemorrhagiae and the routine practice of using a single representative serogroup member in the MAT panel, for example, serovar Copenhageni alone, would have resulted in an underestimation of the seroprevalence. This observation highlights the importance of choosing a multi-serovar panel, including members of the same serogroup, based on established geographic prevalence data. In routine surveillance testing, 1:100 is used as the cut-off titer in diagnostic applications; however, titers as low as 1:10 are generally utilized in reservoir hosts to improve the detection sensitivity (Ellis WA, 1986). 

Another interesting but intriguing finding in this study was that rats, including the ones from which we isolated *L. borgpetersenii* serogroup Ballum, did not show any evidence of an agglutinating antibody response to this serogroup. A similar observation was recorded in rats in one of the comprehensive studies published in the 1980s [25]. In addition, the abovementioned study also reported serogroup Ballum infection in *R. rattus* in high-density populations near garbage, suggesting the influence of ecological factors in the transmission and maintenance of this serogroup. We isolated serogroup Ballum from one *R. norvegicus,* and four *R. rattus* and both species were collected at the same location, suggesting frequent intermingling in Saint Kitts due to the small size of the island and less distinct separation between tropical forest and human-inhabited areas.

The absence of an immune response to a circulating *Leptospira* serogroup can lead to possible false negative results in serological surveys and can also contribute to underestimation of the overall prevalence. The absence of an immune response to *L. borgpetersenii* serogroup Ballum, even in the rats with active infection with this strain, is intriguing but worth speculating. Potential co-evolutionary relationships between host and pathogen species, geographical variations, various ecological factors, including cohabitation of both rat species, mode of transmission, or selection of cell-mediated response over humoral response, or even host-dependent pathogen-specific differential antigen expression can be considered. For example, vertical or pseudovertical transmission and potential immune tolerance could have an immunosuppressive role dampening the immune response [26,27]. In cattle, *L. borgpetersenii* serovar Hardjo infection has been shown to induce a cell-mediated immune response [28]. Therefore, based on our findings in rats, it is logical to hypothesize that *L. borgpetersenii* species may have attributes that direct a cellular response in some hosts, compared to the detectable humoral immune response induced by *L. interrogans*. 

Serogrouping does not have any taxonomical significance but is used as one of the practical preliminary steps in strain characterization. Further differentiation to the basic systematic unit, the serovar, involves laborious and time-consuming cross-agglutination absorption assays (CAAT) and monoclonal antibody-based typing [29,30]. DNA-based techniques have been described to characterize *Leptospira* strains. Whole genome sequencing using the next-generation sequencing methods has become a cost-effective and powerful tool for bacterial strain classification for clinical, biological, and epidemiological investigations. In our study, we focused on a presumptive identification of the isolates at the serogroup level using MAT and then at the species level using WGS. We acknowledge that serovar identification using WGS has not been developed fully, and future work will need to focus on characterizing antigenic regions and other conserved sequences to improve certainty in inferring serovar.

Multiple procedures employed in this study, including culture, WGS, serogrouping, and additional PCRs targeting species-specific sequences, confirmed the occurrence of mixed infections with *L. interrogans* and *L. borgpetersenii* in three rats. Mixed infections have been reported on rare occasions using PCR primers targeting different species [31,32]. Mixed infections in rats were also suggested in a study conducted in Indonesia, using different culture media [33]. To prove the presence of mixed infections in kidneys, and to avoid misinterpretation due to potential laboratory contamination during the isolation steps, we conducted PCR on the original frozen kidney samples, using previously published species-specific primers. Our findings confirmed the evidence of two distinct *Leptospira* spp. present in a few of the rat kidneys. During the lengthy duration required for *Leptospira* culture, conditions may favor the growth of one species over the other, resulting in failure of concurrent growth of infecting strains; therefore, checking for coinfection during the early days of *Leptospira* culture is recommended. 

*Leptospira* isolates are needed for microbiological and epidemiological characterization as well as for diagnostic and vaccine development; however, difficulty in cultivating and isolating *Leptospira* remains one of the major barriers. In our study, we employed frequent subculture from uncontaminated samples and filtration of contaminated samples to allow the unhindered growth and maintenance of bacteria. While most studies have found *Leptospira* serogroup Icterohaemorrhagiae in rats [5,26,34], we believe that these studies have not used diagnostic methods that would facilitate the identification of mixed infections. In our study, WGS analysis separated the genomes of the two co-infecting species. We believe that mixed infections are a likely and under-assessed aspect of *Leptospira* epidemiology and may have important implications for accurate diagnosis, transmission dynamics, and potential disease outcomes. Where infection is highly prevalent in a reservoir population, mixed infections and sequential infection with two pathogenic *Leptospira* species may occur. In incidental hosts, there is some evidence that severe outcomes due to *Leptospira* infection, such as pulmonary hemorrhage syndrome, are immune mediated [35] and the potential role of a priming effect of the initial immune response due to coinfection or sequential infection deserves further studies.

Saint Kitts is a small, surface-limited, insular geographic region, and we are confident that the sample size used in this study is adequate to establish prevalence, based on the previous recommendations reported in this field [36,37]. In summary, the data presented in this study provides useful information to revisit when designing future surveillance studies. Our study, combined with the data gathered from our previous studies, confirms that Saint Kitts is an endemic hotspot for *Leptospira*, with a potentially high transmission risk to humans and animals. Local initiatives to increase public awareness, implement effective rodent control and proper biosecurity measures, and practice vaccination of the livestock and companion animals are suggested to prevent leptospirosis among humans and animals. Our findings also warrant that the choice of diagnostic tests for surveillance or diagnosis will have a significant impact on the prevalence estimates in epidemiological studies as well as on patient management in clinical cases.

## Figures and Tables

**Figure 1 animals-10-00350-f001:**
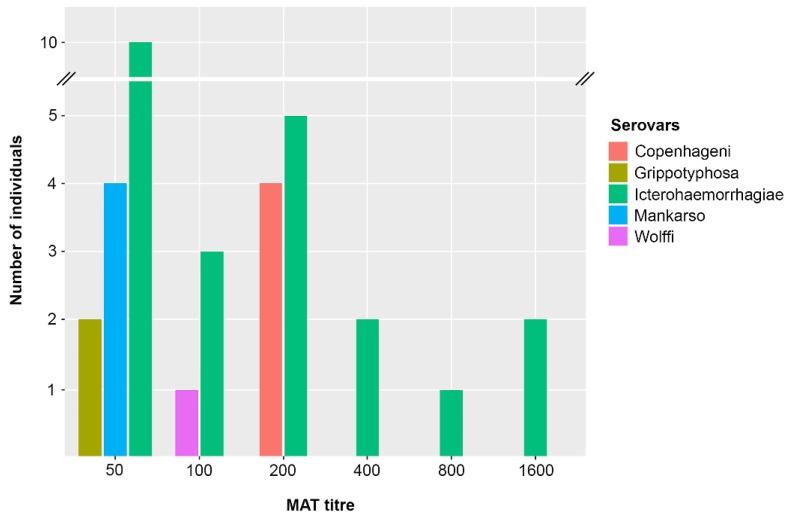
Distribution of the microscopic agglutination test (MAT) titers in rats. The initial starting titer was 1:50.

**Figure 2 animals-10-00350-f002:**
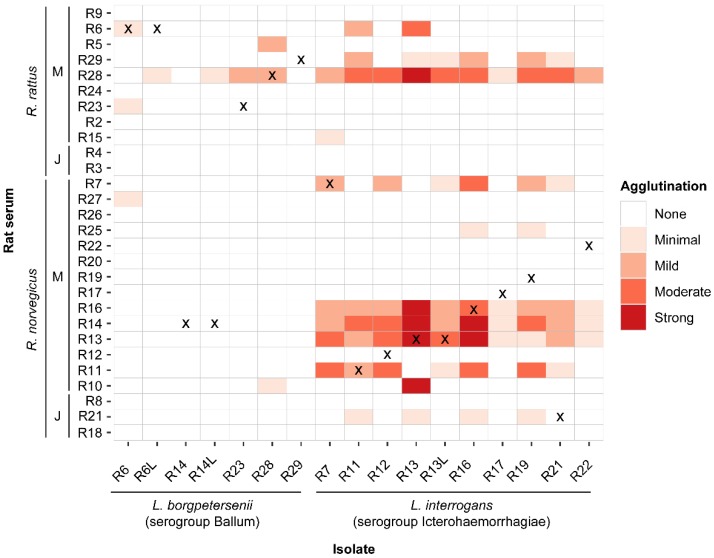
Cross-isolate MAT Assay. MAT response of individual rat serum and rat *Leptospira* isolates from this study. The X indicates the response of an individual rat to its own isolate. M = mature, J = juvenile.

**Table 1 animals-10-00350-t001:** Summary of *Leptospira* detection in rat kidneys.

Species	Number Tested	Number of Positives (Percentage; 95% Confidence Interval)		
		RT-PCR	DFA	Culture
*Rattus* spp.	29	17 (59; 41–77)	11 (38; 20–56)	14 (48; 30–67)
*R. norvegicus*	18	12 (67; 45–89)	9 (50; 27–73)	10 (56; 33–79)
*R. rattus*	11	5 (46; 16–75)	2 (18; 0–41)	4 (36; 8–65)

**Table 2 animals-10-00350-t002:** Characterization of isolates using whole genome sequencing (WGS) and conventional serogrouping.

Isolate Name	WGS ID	Serogrouping Results
R6 (RR)	*L. borgpetersenii*	Ballum (Alexi) ^1^
R6L(RR)	*L. borgpetersenii*	Ballum (Alexi)
R7 (RN)	*L. interrogans*	Ictero/Copenhageni/Mankarso (Canicola)
R11 (RN)	*L. interrogans*	Ictero/Copenhageni/Mankarso (Canicola)
R12 (RN)	*L. interrogans*	Ictero/Copenhageni/Mankarso (Canicola)
R13 (RN)	*L. interrogans*	Ictero/Copenhageni/Mankarso
R13L (RN)	*L. interrogans*	Ictero/Copenhageni/Mankarso
R14 (RN)	*L. borgpetersenii*	Ballum
R14L (RN)	*L. borgpetersenii*	Ballum/Ictero
R16 (RN)	*L. interrogans*	Ictero/Copenhageni/Mankarso
R17 (RN)	*L. interrogans*	Ictero/Copenhageni/Mankarso
R19 (RN)	*L. interrogans*	Ictero/Copenhageni/Mankarso
R21 (RN)	*L. interrogans*	Ictero/Copenhageni/Mankarso
R22 (RN)	*L. interrogans*	Ictero/Copenhageni/Mankarso
R23 (RR)	*L. borgpetersenii*	Ballum
R28 (RR)	*L. borgpetersenii*	Ballum
R29 (RR)	*L. borgpetersenii*	Ballum

^1^ The serovars in parenthesis in the serogrouping results are the isolates with additional mild cross reactivity. Serovars Icterohaemorrhagiae, Copenhageni and Mankarso belong to the same serogroup. RN = *R. norvegicus*; RR = *R. rattus*; Ictero = Icterohaemorrhagiae.

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
