# Peer review of "Detection and Characterization of Leptospira Infection and Exposure in Rats on the Caribbean Island of Saint Kitts"

_animals, 2020, doi:10.3390/ani10020350_

Round 1

Reviewer 1 Report

The manuscript presents novel and important information of one of the most important zoonotic pathogens in the Caribbean. The authors do a great job presenting the data and only few suggestions below.

I think the authors should present a map of St. Kitts with the locations of the rat collection, and description of the areas at least in terms of human/animal populations in the surroundings. This information could be part of the supplementary material, but it is necessary and missing from the paper. Also information of the time of year and year of collection may be of interest. Multiple references for other studies in the areas are presented, but the authors don't compare the serotypes or genotypes that were identified then to what is presented in this paper. Are similar serotypes detected in the multiple animal species? What are the recommendations for vaccination protocols in animals or humans from all these findings?

Author Response

Reviewer 1

The manuscript presents novel and important information of one of the most important zoonotic pathogens in the Caribbean. The authors do a great job presenting the data and only few suggestions below.

Response

Thank you very much for taking the time to review this manuscript. We appreciate your comments and valuable suggestions.

I think the authors should present a map of St. Kitts with the locations of the rat collection, and description of the areas at least in terms of human/animal populations in the surroundings. This information could be part of the supplementary material, but it is necessary and missing from the paper. Also information of the time of year and year of collection may be of interest. Multiple references for other studies in the areas are presented, but the authors don't compare the serotypes or genotypes that were identified then to what is presented in this paper. Are similar serotypes detected in the multiple animal species? What are the recommendations for vaccination protocols in animals or humans from all these findings?

Response

We have included a map and the requested information in a supplementary file (Figure S1, Table S2)

The collection period and the year was already given in the results section. Specific details are included in the new supplementary file (Table S2).

We have included few sentences in the discussion to compare “the serotypes or genotypes that were identified then to what is presented in this paper” in the revised manuscript. Thank you for this excellent suggestion.

We are not comfortable in making recommendations for vaccination protocols from the data gathered.  We have added few lines on vaccination practices in animals on the island.

Reviewer 2 Report

This manuscript reports on an investigation of Leptospira spp. in rats in Saint Kitts. While the direct risk posed by Saint Kitts rats is of relatively minor interest for an international audience, the results are highly valuable scientifically given the island ecological situation and the amount of data alreay available from this island. Hence this study helps understanding more about the role of rats in the eco-epidemiology of leptospirosis in an island settings. In addition the use of multiple testing methods provide important methodological information for future leptospirosis research. This however needs to be detailed more in the manuscript - see below.

The manuscript is overall of good quality and I have mostly minor comments. The cross-isolate assay is to be commended and supports the observation of rats being mostly seronegative for Ballum while being likely infected. 

Abstract:

please indicate the sample size please remove decimal points when presenting percentages, as you have only n=29 at most; this applies to the rest of the manuscript 

Introduction

l63-66: please nuance this statement. The fact that rats are the key reservoir for lepto is not necessarily true for all locations and all serovars l71-73: please develop this part and give the numbers, as the situation in Saint Kitts is important to understand the context of the study and readers may not be familiar. If needed the first paragraph can be reduced as it is mostly general background information about leptospirosis. I would actually suggest to merge both paragraphs and present the local situation at the same time as the disease epidemiology

Methods

sample size calculations l86-88: what level of precision was assumed? the number n=9 is quite low so I suspect it is for detecting Leptospira, which is unlikely to be enough to compare between serovars. I would suggest to redo the calculations to have an idea of the precision reached with the sample size used and the design prevalence l88: please detail the capture as this is important for assessing selection bias. Specifically,  trap model, density or spacing of traps, frequency of checks, bait, local season, location/habitat (e.g; urban, forest, farm...) l109 please define DFM please define the cross-isolate assay and detail what model(s) was/were done; was one model done per serovar or all of them together? it seems more logical to build one model per serovar as the associations may differ for different serovar, e.g. you may expect an age effect for host adapted serovars circulating in an endemic way; or serovars may be different between rat species. what cutoff was used?was a sensitivity analysis done to see the effect of the choice of cutoff? 

Results

My main comment for the results would be to expand some figures and tables to present more information:

table 2: it could be really useful to include all individual data here, at least culture, DFA and PCR results in addition to WGS, as well as data about co-infection presented l193 onwards. This will help understand agreement (or lack of) between different methods. It will also make it easier for the data to be used for meta-analyses for example figure 2: please also indicate the serogroup for the isolates and explain what the numbers for "response" correspond to

Discussion

The discussion is generally well conducted. Discussion on how the findings fit in what is known about the epidemiology of leptospirosis in St Kitts is lacking. For example, recall what serovars are circulating in humans and if contact with rats have been reported as a local risk factor. How would this translate into interventions? 

Author Response

Reviewer 2

This manuscript reports on an investigation of Leptospira spp. in rats in Saint Kitts. While the direct risk posed by Saint Kitts rats is of relatively minor interest for an international audience, the results are highly valuable scientifically given the island ecological situation and the amount of data alreay available from this island. Hence this study helps understanding more about the role of rats in the eco-epidemiology of leptospirosis in an island settings. In addition the use of multiple testing methods provide important methodological information for future leptospirosis research. This however needs to be detailed more in the manuscript - see below.

The manuscript is overall of good quality and I have mostly minor comments. The cross-isolate assay is to be commended and supports the observation of rats being mostly seronegative for Ballum while being likely infected.

Response

Thank you very much for taking the time to review this manuscript. We appreciate your comments and valuable suggestions.

Abstract:

please indicate the sample size please remove decimal points when presenting percentages, as you have only n=29 at most; this applies to the rest of the manuscript

Response

We have made the suggested changes

Introduction

l63-66: please nuance this statement. The fact that rats are the key reservoir for lepto is not necessarily true for all locations and all serovars

Response

Modified this statement

l71-73: please develop this part and give the numbers, as the situation in Saint Kitts is important to understand the context of the study and readers may not be familiar. If needed the first paragraph can be reduced as it is mostly general background information about leptospirosis. I would actually suggest to merge both paragraphs and present the local situation at the same time as the disease epidemiology

Response

Modified

Methods

sample size calculations l86-88: what level of precision was assumed? the number n=9 is quite low so I suspect it is for detecting Leptospira, which is unlikely to be enough to compare between serovars.

I would suggest to redo the calculations to have an idea of the precision reached with the sample size used and the design prevalence

Response

The study was initiated as a pilot prevalence study in rats for the first time on Saint Kitts. We used the recommendations from the pioneers of this field (references 12, 36, 37) to design this study. We are aware that the numbers are not enough to make robust comparisons, however even with low sample size, we are excited that this study provides useful information to consider when designing future studies.

l88: please detail the capture as this is important for assessing selection bias. Specifically, trap model, density or spacing of traps, frequency of checks, bait, local season, location /habitat (e.g; urban, forest, farm...)

 Response

As stated above, the study was started as a basic prevalence study, so we used a convenient sampling strategy. Saint Kitts is a very small island, only 18 miles long and 5 miles wide with indistinct geographic separations. One of the major issues we faced in trapping locations was frequent stealing of traps so we had to use defined locations for trapping. However, a study conducted at a larger scale is still needed to better understand any nuances in prevalence and response. Information requested is provided in the new supplementary table (Table S2).

l109 please define DFM

Response

DFM is defined in line 107

please define the cross-isolate assay and detail what model(s) was/were done; was one model done per serovar or all of them together? it seems more logical to build one model per serovar as the associations may differ for different serovar, e.g. you may expect an age effect for host adapted serovars circulating in an endemic way; or serovars may be different between rat species. what cutoff was used?was a sensitivity analysis done to see the effect of the choice of cutoff?

Response

Our primary goal with the cross-isolate model was to look at differences in host response to serovar, accounting for other host factors such as host maturity and species. Due to sample size limitations, we did not include an interaction term between host species and serovar, but future work with a larger sample size should look at these important factors. We comment on this in the discussion, particularly in reference to potential host adaptation to specific serovars. The statistical analysis section in materials and methods also describes this part.

Results

My main comment for the results would be to expand some figures and tables to present more information:

table 2: it could be really useful to include all individual data here, at least culture, DFA and PCR results in addition to WGS, as well as data about co-infection presented l193 onwards. This will help understand agreement (or lack of) between different methods. It will also make it easier for the data to be used for meta-analyses for example figure 2: please also indicate the serogroup for the isolates and explain what the numbers for "response" correspond to

Response

It is difficult to include individual animal data in the main manuscript file due to space constraints. We have added this information to new supplementary table (S2).

We have also included the serogroup information in the Figure 2 and added a descriptive rather than numerical scale to the figure

.Discussion

The discussion is generally well conducted. Discussion on how the findings fit in what is known about the epidemiology of leptospirosis in St Kitts is lacking. For example, recall what serovars are circulating in humans and if contact with rats have been reported as a local risk factor. How would this translate into interventions?

Response

Rodent control is an essential part of prevention of leptospirosis and other infections. Unfortunately, no study so far has characterized the serovars infecting humans on the island or the risk for zoonosis. We hope that the information from this study will allow the country to take appropriate control measures/intervention strategies.